# A Study Validating the Estimation of Anterior Chamber Depth and Iridocorneal Angle with Portable and Non-Portable Slit-Lamp Microscopy

**DOI:** 10.3390/s21041436

**Published:** 2021-02-19

**Authors:** Eisuke Shimizu, Hiroyuki Yazu, Naohiko Aketa, Ryota Yokoiwa, Shinri Sato, Junichiro Yajima, Taiichiro Katayama, Rio Sato, Makoto Tanji, Yasunori Sato, Yoko Ogawa, Kazuo Tsubota

**Affiliations:** 1Department of Ophthalmology, Keio University School of Medicine, Tokyo 160-8582, Japan; g.h.yazu@gmail.com (H.Y.); nao.nao.pao.pao@gmail.com (N.A.); shinri.sato259@keio.jp (S.S.); bdkog460@gmail.com (J.Y.); taiichiro.katayama@gmail.com (T.K.); rio.sato406@gmail.com (R.S.); tanji.makoto@gmail.com (M.T.); yoko.z7@keio.jp (Y.O.); tsubota@z3.keio.jp (K.T.); 2OUI Inc., Tokyo 160-0022, Japan; ryota.yokoiwa@gmail.com; 3Yokohama Keiai Eye Clinic, Kanagawa 240-0065, Japan; 4Department of Ophthalmology, Tsurumi University School of Dental Medicine, Kanagawa 230-8501, Japan; 5Department of Preventive Medicine and Public Health, School of Medicine, Keio University, Tokyo 160-8582, Japan; yasunori.sato@keio.jp

**Keywords:** smart eye camera, slit-lamp microscope, anterior chamber depth, trabecular-iris angle, anterior-segment optical coherence tomography

## Abstract

This study assessed the anterior chamber depth (ACD) and iridocorneal angle using a portable smart eye camera (SEC) compared to the conventional slit-lamp microscope and anterior-segment optical coherence tomography (AS-OCT). This retrospective case-control study included 170 eyes from 85 Japanese patients. The correlation between the ACD evaluations conducted with the SEC and conventional slit-lamp was high (r = 0.814). The correlation between the Van-Herick Plus grade obtained using two devices was also high (r = 0.919). A high kappa value was observed for the Van-Herick Plus grading (Kappa = 0.757). A moderate correlation was observed between the ACD measured using AS-OCT and the slit-lamp image acquired with the conventional slit-lamp microscope and SEC (r = 0.609 and 0.641). A strong correlation was observed between the trabecular-iris angle (TIA) measured using AS-OCT and Van-Herick Plus grade obtained with the conventional slit-lamp microscope and SEC (r = 0.702 and 0.764). Strong correlations of ACD evaluation and high kappa value of the Van-Herick Plus grading indicated the adequate subjective assessment function of the SEC. Moderate correlations between the ACD objective measurement and evaluation and strong correlation between the TIA and Van-Herick Plus grade suggested the good objective assessment function of the SEC. The SEC demonstrated adequate performance for ACD evaluation and angle estimation.

## 1. Introduction

The slit-lamp microscope is the most commonly used diagnostic instrument in daily ophthalmological practice [1]. One of the purposes of slit-lamp examination is the depiction of the anterior chamber angle to screen for possible pharmacologic pupillary dilation [2]. The direct focal illumination method, one of the various illumination methods, entails the use of a thin slit-beam for the evaluation of the anterior chamber depth (ACD) and iridocorneal angles [3]. Generally, a slit-lamp microscope is installed in the hospital’s ophthalmology department and used by trained ophthalmologists for subjective evaluations [4]. Therefore, patients are required to visit the medical center for ophthalmologic examinations. Although some types of portable slit-lamp microscopes exist, these instruments lack a recording function [5]. Moreover, these instruments are basically disconnected from the internet to safeguard patients’ medical information. Some smartphone-based slit-lamp devices have already been developed, but these devices have not yet become common [5]. The recently developed state-of-the-art anterior-segment optical coherence tomography (AS-OCT) enables one to provide the optic pathology in real time using low-coherence interferometry [6,7]. It has been developed for noninvasive cross-sectional imaging in biological tissue including anterior chamber [6,8]. Enabled to quantitatively measure various anterior chamber parameters as corneal curvature radius, corneal thickness, corneal higher-order aberrations, and trabecular-iris angle (TIA) [9,10,11,12,13,14]. However, the AS-OCT device (which makes objective measurements) is also unmovable, similar to the portability issue associated with the conventional slit-lamp microscopes [5]. Thus, we invented a portable and recordable slit-lamp device, named the smart eye camera (SEC), to address the issues associated with immovability and recordability. The SEC converts the light of the smartphone to the essential light needed for the ophthalmological diagnosis. We previously demonstrated its diagnostic function for cataracts in humans and the dry eye disease mouse model [5,15,16]. However, no study has investigated the ACD evaluation and angle estimation function of the portable and recordable slit-lamp device. Therefore, we conducted a verification study to assess the ability of the new slit-lamp SEC device for ACD evaluation and angle estimation. Evaluation with the conventional slit-lamp microscope is subjective, because of grading by ophthalmologists, since there is the possibility of interobserver variations. Hence, the corresponding parameters were measured objectively using AS-OCT. We evaluated the diagnostic function of the SEC by comparing the subjective grades obtained using conventional slit-lamp microscopy and objective measurement obtained using AS-OCT in this study.

## 2. Materials and Methods

### 2.1. Ethics Approval

The retrospective study adhered to the tenets of the Declaration of Helsinki. The clinical trial was registered with the University Hospital Medical Information Network Center (UMIN-CTR: UMIN000040321). The study protocols were approved by the institutional ethics review board (IRB) of Keio University School of Medicine, Tokyo, Japan (IRB No. 20200021). Written informed consent was obtained from all patients via an agreement document.

### 2.2. Study Design

A fellow (T.K.) and resident (R.S.) reviewed the database (Claio, FINDEX Inc., Tokyo, Japan) of the Department of Ophthalmology, Keio University School of Medicine, to screen for eligible patients.

A fellow or resident recorded the eye images by SEC for the prediagnosis and an attending physician used the conventional slit-lamp microscope for confirming the diagnosis. The inclusion criteria were as follows: (1) Japanese men and women (above 20 years of age), (2) patients who underwent slit-lamp examination with both conventional slit-lamp microscopy and SEC, and (3) patients who underwent AS-OCT examinations on the same visiting date. The exclusion criteria were as follows: (1) pseudophakic eyes, (2) aphakic eyes, and (3) eyes with mydriasis because it could possibly alter the spherical curve of the lens. Sample size calculation yielded a required sample size of 146 eyes (details are provided in the Statistical and Data Analysis Section). Therefore, we enrolled 170 eyes from 85 Japanese patients between August and September 2020. A fellow (T.K.) and resident (R.S.) extracted all the data from the electronic medical records (EMR; HOPE EGMAIN-GX, Fujitsu Limited, Tokyo, Japan and Claio, FINDEX Inc., Tokyo, Japan). The extracted data were randomized by a random digit panel to avoid any bias and classified into the following datasets: (I) slit-lamp photographs obtained using the conventional slit-lamp microscope, (II) slit-lamp videos obtained using the SEC, and (III) the objective anterior chamber value in the form of AS-OCT findings (ACD and TIA). An attending physician (S.S.) subjectively evaluated all the (I) slit-lamp images and (II) SEC videos after data curation in a masked manner to avoid bias and determined whether the depth of the anterior chamber was deep or shallow subjectively. Next, the physician evaluated the Van-Herick Plus grade by focusing on the inferior corneal angle. To facilitate performance of the two devices, the correlation analysis and reproducibility calculation were performed between the conventional slit-lamp microscope and the SEC (the depth of the anterior chamber and Van-Herick Plus grading). Moreover, to confirm similar performance between the two devices, the correlation analysis was performed between (III) the objective anterior chamber value from AS-OCT (ACD or TIA) and either subjective depth of the anterior chamber or Van-Herick Plus grading.

### 2.3. Diagnostic Instruments and Slit-Lamp Examinations

Several conventional slit-lamp microscopes were used for anterior chamber image acquisition: the SL130 (Carl Zeiss AG, Oberkochen, Germany), 700GL (Takagi, Nagano, Japan), and SL-D301 (TOPCON, Tokyo, Japan). These slit-lamp microscopes are non-portable and are normally used in the conventional examination room (Figure 1). Each anterior segment image was recorded by the external camera installed in the slit-lamp microscopes (THD-23FHD; IKEGAMI TSUSHINKI CO., LTD., Tokyo, Japan. MCC-500MD; Sony Business Solutions Corporation, Tokyo, Japan, SP-321 FREX HD; JFC Sales Plan Co., Ltd. Tokyo, Japan) and converted to a JPEG file. The resolution of the photographs was 1920 pixels × 1200 pixels per inch. The images obtained with the portable SEC with a recording function (OUI Inc., Tokyo Japan) were verified (Figure 1). The SEC has a function that converts the light source from the smartphone into a 1-mm thin beam, allowing it to fall on the crystalline lens. Moreover, the SEC has a convex lens that magnifies the reflected light to a degree sufficient for ocular diagnosis [5,15,16]. The iPhone 7, 8, or SE (Apple Inc., Cupertino, CA, USA) was used to record the ocular video. The resolution of the video was 1080p, with a frame rate of 30 frames per second. The direct focal illumination method was used for both examination tools with the slit-light angle fixed to 40° and 1-mm slit-beam. We could not document the order of examination with the two instruments due to the nature of the retrospective study. However, the recording conditions (e.g., brightness, temperature, humidity, etc.) were maintained consistently to prevent the influence of any environmental factors. Both tools were used in the same dark examination area (30-75 lux. JIS Z 9110-2010: General rules of recommended lighting levels). The depth of the anterior chamber was subjectively evaluated from each conventional slit-lamp and SEC images from the magnified, 3D images structured by the slit-light beam [17]. Van-Herick Plus grading was also performed subjectively focusing on the inferior corneal angle image. It is the modified criteria for the golden standard “Van-Herick grading” method for both peripheral ACD and angle assessment [18]. Van-Herick grading uses nasal or temporal (0 or 180°) images. Controversially, Van-Herick Plus grading uses inferior (270°) angle. Inferior peripheral ACD to peripheral corneal thickness ratio was estimated from the magnified photos. The subjects were divided into four groups to judge the peripheral angle: 1 (<1/4), 2 (1/4 to 1/2), 3 (>1/2 to 1), and 4 (>1) [18].

### 2.4. Anterior-Segment Optical Coherence Tomography

Corneal/AS-OCT was performed using the CASIA2 Advance system (Tomey, Nagoya, Japan: wavelength: 1310 nm; resolution: <10 µm; scan ratio: 50,000 per seconds). All participants were examined by several certified optometrists until at least two sets of images of sufficient quality were recorded. All the OCT scans were centered on the pupil and acquired along the vertical axis (superior and inferior angles of 90° and 270°, respectively), using the CASIA system’s standard anterior-segment scan protocol [19]. The ACD was measured as the perpendicular distance between the apex of the posterior corneal margin (corneal endothelium) and apex of the crystalline lens using AS-OCT [19,20]. The TIA was measured with the apex in the iris angle recess and the arms of the angle passing through a point on the trabecular meshwork 500 μm from the scleral spur (SS) and the point on the iris perpendicularly (inferior corneal angle of 270°) using AS-OCT [20]. These ACD and TIA-related sites were automatically identified by the AS-OCT (Figure 2). During data curation, a corneal specialist (N.A.) and corneal fellow (T.K.) reconfirmed all AS-OCT images to ensure that adequate digitalization and recognition of the corneal and lens surfaces by the automated software were adequate for the objective findings in our study.

### 2.5. Statistical and Data Analysis

All data were analyzed using SPSS (ver. 25; International Business Machines Corporation, Armonk, NY, USA). The sample size calculation was performed based on the previous data [5]. Difference between two independent means and correlation between the groups used to calculate an effect size of 0.31, statistical power of 0.95, and significance level of 0.05. A total sample size of 146 was determined to be adequate. Spearman’s correlation coefficient was used to evaluate the correlation between the subjective depth of the anterior chamber and Van-Herick Plus grades obtained using conventional slit-lamp microscopy and SEC. A weighted kappa coefficient with a 95% confidence interval (CI) was used to evaluate the reproducibility of the Van-Herick Plus grade obtained with the conventional slit-lamp microscope and SEC. The correlation analysis between the objective AS-OCT findings and conventional slit-lamp or SEC was conducted using Spearman’s correlation coefficient. The tests for significance were accompanied by the 95% CI. Data were expressed as the mean ± standard deviation.

## 3. Results

### 3.1. Patient Characteristics

This study enrolled 170 eyes from 85 Japanese patients (45 men and 40 women). The participant’s mean age was 44.393 ± 19.798 years (range: 20–96 years). The size of a single ocular photograph acquired with the conventional slit-lamp and evaluated by the ophthalmologist was 696.364 ± 103.571 kilobyte. The length of the video recorded by the SEC was 10.940 ± 2.464 s (Table 1). The ACD was measured successfully using AS-OCT in 164 eyes. The average ACD was 2.912 ± 0.391 mm (range: 1.374–3.613 mm), 2.911 ± 0.358 mm (range: 1.940-3.613 mm), and 2.912 ± 0.374 mm in the right, left, and both eyes respectively (Table 1). AS-OCT successfully measured the TIA in 148 eyes. The average TIA was 39.792 ± 12.984° (range: 10.1–66.9°), 39.845 ± 13.546° (range: 10.4–72.7° degrees), and 39.818 ± 13.223° in the right, left, and both eyes respectively (Table 1). A comparison of the visual characteristics of the same patient acquired using the conventional slit-lamp microscope and SEC is presented in Figure 3.

### 3.2. Correlation and Reproducibility between Conventional Slit-Lamp Microscopy and SEC

A total of 148 eyes were included in the correlation analysis. The correlation coefficient between the ACD measured with the conventional slit-lamp and SEC was high in the right, left, and both eyes (r = 0.828, 95% CI: 0.786–0.885; r = 0.792 95% CI: 0.696–0.860; and r = 0.814, 95% CI: 0.756–0.860; respectively; Table 2). Moreover, the correlation coefficient between the conventional slit-lamp and SEC for Van-Herick Plus grading was also high in the right, left, and both eyes (r = 0.914, 95% CI: 0.870–0.943; r = 0.924, 95% CI: 0.885–0.950; and r = 0.919, 95% CI: 0.892–0.940; respectively; Table 2). Furthermore, a high kappa value was observed between conventional slit-lamp and SEC for the Van-Herick Plus grade (Kappa = 0.757 95% CI 0.749–0.765. Table 3).

### 3.3. Correlation between AS-OCT Findings and Conventional Slit-Lamp or SEC

A moderate correlation coefficient was observed between ACD measured by AS-OCT and slit-lamp image with the conventional slit-lamp (r = 0.609, 95% CI: 0.504–0.696, 146 eyes; Table 4). Similarly, a moderate correlation coefficient was observed between ACD measured with AS-OCT and that obtained with the slit-lamp image with SEC (r = 0.641, 95% CI: 0.543–0.722, 163 eyes; Table 4). On the other hand, a strong correlation coefficient was observed between the TIA measured using AS-OCT and Van-Herick Plus grade obtained using the conventional slit-lamp (r = 0.702, 95% CI: 0.617–0.771, 131 eyes; Table 4). Similarly, a strong correlation coefficient was observed between the TIA measured using AS-OCT and the Van-Herick Plus grade obtained with the SEC (r = 0.764, 95% CI: 0.693–0.820, 147 eyes; Table 4).

## 4. Discussion

The present study aimed to assess the efficacy of ACD evaluation and anterior chamber angle estimation by the new slit-lamp SEC device. We evaluated the function of the SEC by comparing it with the measurements obtained with subjective conventional slit-lamp microscopy and objective AS-OCT. The size of the current study sample was calculated on the basis of a similar validation study [5]. We decided to screen the information stored in the database of Keio University School of Medicine in order to obtain a study population of the desired size. Our appropriate dataset included a sufficient number of eyes (170 eyes) from Japanese patients only with a balanced age (44.393 ± 19.798 years), and sex (45 men and 40 women, Table 1) distribution. Moreover, values of the ACD and TIA obtained using AS-OCT were also well balanced. A previous study in another Asian population showed similar results for the ACD (2.85 ± 0.27 mm, range: 2.42-3.48 mm) [21]. Moreover, a study conducted to determine the normative references in another Asian population showed extremely similar results for the anterior chamber angle (38.3 ± 16.3°) [22]. Therefore, a cohort of our dataset was thought to be eligible for further discussion.

In the current study, we assessed the function of the SEC by validating its inter-instrument agreement and reproducibility using a comparison with the conventional slit-lamp microscope. A significantly high correlation was observed between slit-light ACD evaluation with the conventional slit-lamp microscope and SEC (r = 0.814, Table 2). Moreover, ACD measurement using the AS-OCT and the slit-light ACD evaluation using the conventional slit-lamp demonstrated a moderate correlation (r = 0.609, Table 4). Interestingly, a similarly moderate correlation was observed between the AS-OCT measurements and the slit-lamp ACD evaluations by the SEC (r = 0.641, Table 4). The ACD evaluated by the slit-light is subjective, while the ACD measured by AS-OCT is objective. Our results exhibited the same degree of correlation, which suggests sufficient agreement between the SEC and the conventional slit-lamp microscope for ACD evaluation. Moreover, there was a significantly high correlation between the Van-Herick Plus grades obtained by the conventional slit-lamp microscope and SEC (r = 0.919, Table 2). Furthermore, a strong correlation (r = 0.702; Table 4) was observed between the TIA measured by the AS-OCT and the Van-Herick Plus grade obtained with a conventional slit-lamp. Intriguingly, a similar correlation was also observed between the TIA measurements and Van-Herick Plus grade obtained using the SEC (r = 0.764, Table 4). The Van-Herick Plus grade obtained using slit-light evaluation is subjective [18], while TIA measurement with AS-OCT is objective. The similarity in the degree of correlation suggests adequate agreement between the SEC and the conventional slit-lamp microscope for the anterior chamber angle estimation. Moreover, a high kappa value was recorded for the Van-Herick Plus grade obtained with the conventional slit-lamp and SEC (Kappa = 0.757, Table 3), which suggests good reproducibility. Hence, a high kappa value between the Van-Herick Plus grades obtained with the two devices and strong correlation between the TIA and the Van-Herick Plus grade evaluated by the SEC suggests that the SEC has sufficient utility for ACD evaluation and angle estimation.

Several studies have demonstrated the efficacy of smartphone-based slit-lamp devices. Moreover, several studies have indicated that an external convex lens enables adjustment of the magnification to the eyes [23,24]. Recently, Hu S, et al. demonstrated the efficiency of smartphone-based slit-lamps for cataract screening. They used an external light source and the camera function of the smartphone to record intricately detailed images of the mydriasis cataract [25]. Moreover, Pujari A, et al. demonstrated a method for photographing the iridocorneal angle using the latest smartphone and external slit-lamp biomicroscope [26]. Finally, Chiong et al. invented the three-dimensionally (3D) printed smartphone attachment, which uses the smartphone’s light source and camera function [27]. However, they did not succeed in creating a slit-light source using the light source of the smartphone. The SEC utilizes the combination of a thin slit and cylindrical lens, which enables the conversion of the light source of the smartphone into a slit-light beam thin enough to enter the pupil and enable diagnosis of cataracts [5]. There exists a device that uses an external light source to create thin slit-light and subsequently expose it to the eyes [25]. However, these previous studies did not succeed in converting the smartphone’s original light to a thin slit-light source. In summary, to the best of our knowledge, there is no prior published evidence of the use of a smartphone’s light to create a slit-light source for the evaluation of ACD and estimate the anterior chamber angle. Moreover, the SEC was also 3D printed using polyamide resin, which saves the cost of production [27].

Our study had several limitations. First, although we did confirm the subjective findings obtained with slit-lamp photography, we did not perform any gonioscopic evaluation using three or four mirror goniolens. Gonioscopy is the most traditional method used to estimate the angle width using a gonioscope and slit-lamp microscope [28]. Moreover, Scheie, Shaffer, and Spaeth grading system using a gonioscope are currently the gold standard for the diagnosis and classification of angle evaluation in clinical settings [18,29,30,31]. In the current study, we modified the angle evaluation method using the Van-Herick Plus grading, which is reported to be a simple and relatively accurate technique for both ACD and angle assessment [18]. Moreover, the Van-Herick Plus grade showed a correlation with the anterior chamber angle estimated by AS-OCT [18]. Furthermore, other studies have indicated that AS-OCT measurements are highly correlated with the use of gonioscopy and may possibly be used to assess angle closure risk using non-contact methods [32]. The Van-Herick Plus grading system is able to mimic the utility, accuracy, or value of the gonioscopy method; however, it cannot replace the current golden standard as the Van-Herick grading system and several gonioscopic evaluations, which is the definitive method for diagnosis. Therefore, we believe that SEC is suitable for screening risky patients. Moreover, further study is needed to directly evaluate the iridocorneal angle using gonioscope and SEC. Second, this was a retrospective case-control study. Thus, its objective needed to be selected appropriately. Therefore, we extracted the information of every patient, who underwent assessment between August and September 2020 and confirmed the absence of any deviation in the patients’ age and sex because these factors reported to be responsible for the risk of the bias in the ACD or angle differences. Third, we enrolled only Japanese patients. Differences in ethnicity are reported to be risk factors for angle closure due to occludable or/and narrow angles [33,34]. Moreover, Wang et al. demonstrated the difference in the ACD and risk in occludable angles between Caucasians and Asians [35]. This study evaluated several parameters of the anterior portion of the eye using two different devices. Therefore, the differences in ethnicity should not be a substantial limitation. However, future studies with ethnically diverse populations are needed. Fourth, only moderate correlation was observed between subjective SEC analysis and objective AS-OCT analysis (Table 4). Correlation coefficient has a similar value in conventional slit-lamp and SEC (r = 0.609, and r = 0.641). Therefore, it is considerable to increase accuracy, however future studies with more numbers of the cases are needed to gain its precision. Despite these limitations, no other study has demonstrated the utility of a portable and recordable slit-lamp device that utilized only smartphone-based functions. However, prospective studies are needed to verify the effect of the patient’s age, gender, and race.

The availability of ophthalmological examination is limited, especially in developing countries, owing to the lack of expensive medical instruments. Moreover, portable and light weight tools are available in disaster medicine, isolated areas, and visiting care. The SEC is not only portable and recordable but is also easy and inexpensive to manufacture and works in conjunction with a smartphone. Therefore, the SEC has potential as a screening tool for ACD and angle estimation that can improve the availably and enlighten the importance of the ophthalmological examination worldwide. For example, in the population attributable risk percentage (PAR%), which addresses the problem of whether a disease is a public health problem [36], primary angle closure has a three times higher PAR% compared to primary open angle glaucoma [37]. Moreover, wireless smartphones are connected globally, which makes the SEC compatible with telemedicine and remote medicine. Furthermore, ophthalmological examinations are based on imaging diagnosis. Thus, artificial intelligence-based diagnosis facilitated by big data collection could be feasible in the future. Our study demonstrated the utility of a new slit-lamp device, i.e., the SEC, for measuring the ACD and estimating the anterior chamber angle compared to the conventional subjective and objective methods. However, the study design had some limitations, such as the lack of gonioscopic evaluation and inclusion of patients from only one ethnicity in the study population. Hence, further prospective studies are necessary to overcome the limitations of the current study.

## Figures and Tables

**Figure 1 sensors-21-01436-f001:**
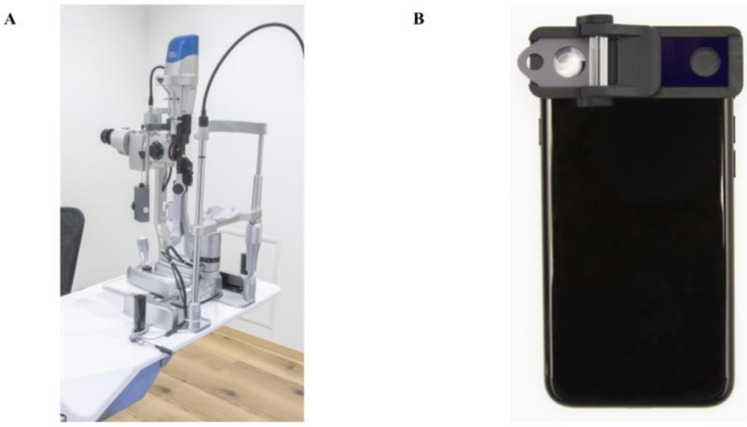
Photographs of the conventional slit-lamp microscope and the smart eye camera (SEC). (**A**) Appearance of the conventional slit-lamp microscope (700GL; Takagi, Nagano, Japan). External camera is installed inside (THD-23FHD; IKEGAMI TSUSHINKI CO., LTD., Tokyo, Japan). (**B**) Appearance of the smart eye camera (OUI Inc., Tokyo Japan). The attachment includes a thin slit and a cylinder lens to converts the light source from the smartphone into a 1-mm thin beam.

**Figure 2 sensors-21-01436-f002:**
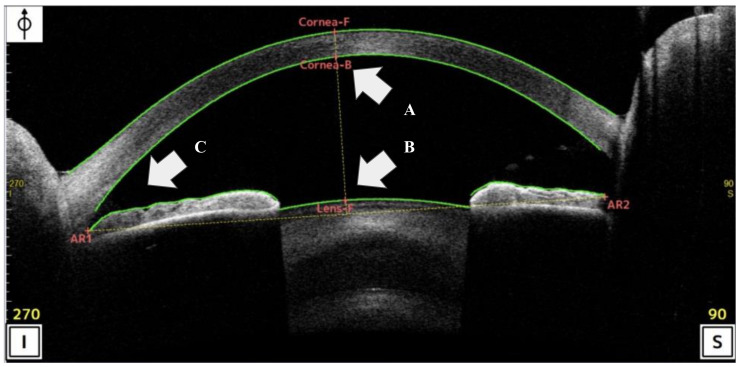
Scan of the analysis interface in anterior-segment optical coherence tomography (AS-OCT) (CASIA2 Advance system: Tomey, Nagoya, Japan). (**A**,**B**) Anterior chamber depth (ACD) was define the distance between the apex of the posterior corneal to apex of the crystalline lens using AS-OCT. (**C**) Trabecular-iris angle (TIA) was measured with the apex in the iris angle recess and the arms of the angle passing through a point on the trabecular meshwork 500 μm from the SS and the point on the iris.

**Figure 3 sensors-21-01436-f003:**
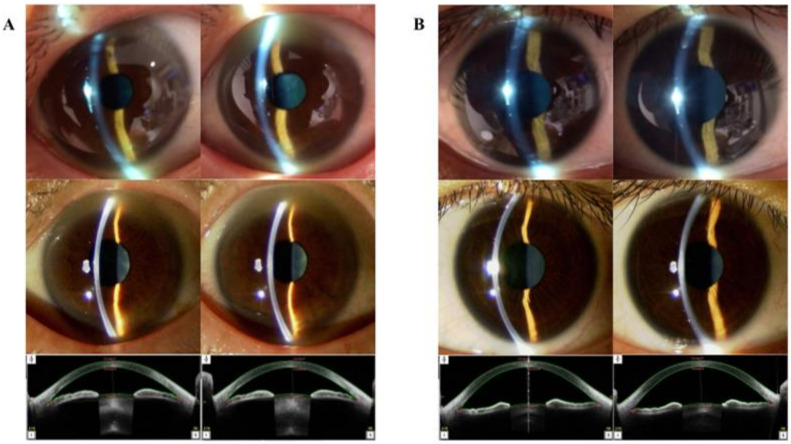
Case presentation. (**A**) Ophthalmological images of a 62-year-old Asian man. The upper-tier images were obtained using the smart eye camera (AC: shallow/shallow, Van-Herick Plus grade: 3/3). The middle-tier images were obtained using conventional slit-lamp microscopy (AC shallow/shallow, Van-Herick Plus grade: 3/3). The bottom-tier images were obtained using AS-OCT (ACD: 2.481/2.484 mm, TIA: 21.3/20.4°). (**B**) Ophthalmological images of a 32-year-old Asian woman. The upper-tier images were obtained using the smart eye camera (AC: deep/deep, Van-Herick Plus grade: 4/4). The middle-tier images were obtained using the conventional slit-lamp (AC deep/deep, Van-Herick Plus grade: 4/4). The bottom-tier images were obtained using AS-OCT (ACD: 3.415/3.388 mm, TIA: 66.7/61.9°). AS-OCT: anterior-segment optical coherence tomography, ACD: anterior chamber depth, TIA: trabecular-iris angle.

**Table 1 sensors-21-01436-t001:** Patients’ characteristics.

Cases, *n*	85		
Eye, *n*	170		
Age	44.393	±	19.798
Sex, *n*			
Male	45		
Female	40		
Conventional image size, KB	696.364	±	103.571
Video length/eye, seconds	10.940	±	2.464
Anterior-Segment Optical Coherence Tomography			
ACD, mm			
Right eye	2.912	±	0.391
Left eye	2.911	±	0.358
Both eyes	2.912	±	0.374
TIA, degrees			
Right eye	39.792	±	12.984
Left eye	39.845	±	13.546
Both eyes	39.818	±	13.223

KB, kilobyte; ACD, anterior chamber depth; TIA, trabecular-iris angle.

**Table 2 sensors-21-01436-t002:** Correlation between conventional slit-lamp microscopy findings and SEC findings.

	*n*	R*	95% CI	
ACD				
Right eye	74	0.828	0.786	0.885
Left eye	74	0.792	0.696	0.860
Both eyes	148	0.814	0.756	0.860
Van-Herick Plus grade				
Right eye	74	0.914	0.870	0.943
Left eye	74	0.924	0.885	0.950
Both eyes	148	0.919	0.892	0.940

SEC, Smart Eye Camera; ACD, anterior chamber depth; CI, confidence interval; * Spearman’s rank correlation coefficient.

**Table 3 sensors-21-01436-t003:** Reproducibility of the Van-Herick Plus grade.

		Conventional Slit-Lamp			
	Grade	1	2	3	4
SEC	1	4	1	0	0
	2	2	10	1	0
	3	1	8	27	1
	4	0	0	6	87
Kappa		0.757			
95% CI		0.749		0.765	

SEC, Smart Eye Camera; CI, confidence interval.

**Table 4 sensors-21-01436-t004:** Correlation between the AS-OCT findings and conventional slit-lamp or SEC findings.

	*n*	R*	95% CI	
ACD versus Slit-lamp image				
Conventional slit-lamp	146	0.609	0.504	0.696
SEC	163	0.641	0.543	0.722
TIA versus Van-Herick Plus grading				
Conventional slit-lamp	131	0.702	0.617	0.771
SEC	147	0.764	0.693	0.820

AS-OCT, anterior-segment optical coherence tomography; SEC, Smart Eye Camera; CI, confidence interval; ACD, anterior chamber depth; TIA, trabecular-iris angle; * Spearman’s rank correlation coefficient.

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
