# Peer review of "A Study Validating the Estimation of Anterior Chamber Depth and Iridocorneal Angle with Portable and Non-Portable Slit-Lamp Microscopy"

_sensors, 2021, doi:10.3390/s21041436_

Round 1

Reviewer 1 Report

The paper approaches the estimation of anterior chamber depth and iridocorneal angle. A classical, fixed slit-lamp is used, as well as a portable one, i.e. the "Smart Eye Camera (SEC)" invented by the authors. This is a valuable approach. Anterior-segment optical coherence tomography (AS-OCT) is also used to validate the results obtained with the invented device. The paper is well-written, and the study is sound, therefore in the opinion of this reader it can be accepted for publication, with some improvements, as suggested bellow.

1) The English is in general fine, although it should still be polished.

2) The Intro should include aspects of the validation techniques, especially OCT, as some readers may not be familiar with them. The same in the Materials and Methods section. References should be provided for the readers, including

3) The seminal work on OCT

  1. Huang, E. A. Swanson, C. P. Lin, J. S. Schuman, W. G. Stinson, W. Chang, M. R. Hee, T. Flotte, K. Gregory, C. A. Puliafito, J. G. Fujimoto, „Optical coherence tomography,” Science 254(5035), 1178-1181, 1991.

4) recent review(s) on OCT for the eye

Drexler, W., Fujimoto, J. G. State-of-the-art retinal optical coherence tomography. Progress in Retinal and Eye Research 2008;27(1);45–88.

5) for cornea imaging, the state-of-the-art 2 microns axial and lateral resolution Gabor-Domain Optical Coherence Microscopy:

  1. Cogliati, C. Canavesi, A. Hayes, P. Tankam, V.-F. Duma, A. Santhanam, K. P. Thompson, and J. P. Rolland, MEMS-based handheld scanning probe with pre-shaped input signals for distortion-free images in Gabor-Domain Optical Coherence Microscopy, Optics Express 24(12), 13365-13374 (2016).

6) In Fig. 1 please mark (a) and (b) and discuss the two figs accordingly in the text.

7) In general, please adhere to the template of the journal. The manuscript should be checked from this point of view.

8) Details on the AS-OCT system should be given. Is it an experimental or a commercial one? For the latter, please provide manufacturer. In any case parameters must be indicated, including center wavelength, resolution, acquisition speed, etc.

9) Line 165, “The sample size was calculated according to a previous study”. Please for all such aspects provide all the necessary details for the reader in the text. The paper must be stand-alone.

10) Some of the notations in the text may be spared, to make it more readable. Please check and keep only relevant ones.

11) Very good and well-thought Discussion section, highlighting both novelty and limitations.

Author Response

Reviewer1

The paper approaches the estimation of anterior chamber depth and iridocorneal angle. A classical, fixed slit-lamp is used, as well as a portable one, i.e. the "Smart Eye Camera (SEC)" invented by the authors. This is a valuable approach. Anterior-segment optical coherence tomography (AS-OCT) is also used to validate the results obtained with the invented device. The paper is well-written, and the study is sound, therefore in the opinion of this reader it can be accepted for publication, with some improvements, as suggested bellow.

Comment. Thank you for the various novel comments. We appreciate the reviewer for the kind instructions. Specific comment and changes are written below.

1) The English is in general fine, although it should still be polished.

Comment 1. Thank you for the novel indication. We revised our manuscript using professional English service. Moreover, we had attached certificate of editing.

2) The Intro should include aspects of the validation techniques, especially OCT, as some readers may not be familiar with them. The same in the Materials and Methods section. References should be provided for the readers, including

3) The seminal work on OCT

  1. Huang, E. A. Swanson, C. P. Lin, J. S. Schuman, W. G. Stinson, W. Chang, M. R. Hee, T. Flotte, K. Gregory, C. A. Puliafito, J. G. Fujimoto, „Optical coherence tomography,” Science 254(5035), 1178-1181, 1991.

4) recent review(s) on OCT for the eye

Drexler, W., Fujimoto, J. G. State-of-the-art retinal optical coherence tomography. Progress in Retinal and Eye Research 2008;27(1);45–88.

5) for cornea imaging, the state-of-the-art 2 microns axial and lateral resolution Gabor-Domain Optical Coherence Microscopy:

  1. Cogliati, C. Canavesi, A. Hayes, P. Tankam, V.-F. Duma, A. Santhanam, K. P. Thompson, and J. P. Rolland, MEMS-based handheld scanning probe with pre-shaped input signals for distortion-free images in Gabor-Domain Optical Coherence Microscopy, Optics Express 24(12), 13365-13374 (2016).

Comment 2-5. Thank you for the novel comment. We revised our manuscript in order to describe more about the OCT technique.

LINE: 52-58

The recently developed state-of-the-art anterior-segment optical coherence tomography (AS-OCT) enables to provide the optic pathology in real time using low-coherence interferometry [32, 33]. It has been developed for noninvasive cross-sectional imaging in biological tissue including anterior chamber [32, 34]. Enabled to quantitatively measure various anterior chamber parameters as corneal curvature radius, corneal thickness, corneal higher-order aberrations, and trabecular-iris angle (TIA) [6-11].

6) In Fig. 1 please mark (a) and (b) and discuss the two figs accordingly in the text.

Comment 6. Thank you for the novel point. We revised figure legend.

LINE: 145-149

Figure 1. Photographs of the conventional slit-lamp microscope and the SEC. (A) Appearance of the conventional slit-lamp microscope (700GL; Takagi, Nagano, Japan). External camera is installed inside (THD-23FHD; IKEGAMI TSUSHINKI CO., LTD., Tokyo, Japan). (B) Appearance of the Smart Eye Camera (OUI Inc, Tokyo Japan). The attachment includes a thin slit and a cylinder lens to converts the light source from the smartphone into a 1-mm thin beam.

7) In general, please adhere to the template of the journal. The manuscript should be checked from this point of view.

Comment 7. Thank you for the kind indication. We revised our manuscript using template. Moreover, we had used editing service to collect the formatting.

8) Details on the AS-OCT system should be given. Is it an experimental or a commercial one? For the latter, please provide manufacturer. In any case parameters must be indicated, including center wavelength, resolution, acquisition speed, etc.

Comment 8. Thank you for the novel comment. The AS-OCT is already in commercial. We added the details in the manuscript.

LINE: 153-154

Corneal/AS-OCT was performed using the CASIA2 Advance system (Tomey, Nagoya, Japan: Wavelength; 1,310 nm: Resolution; < 10µm: Scan ratio; 50,000 per seconds).

9) Line 165, “The sample size was calculated according to a previous study”. Please for all such aspects provide all the necessary details for the reader in the text. The paper must be stand-alone.

Comment 9. Thank you for the novel comment. The AS-OCT is already in commercial. We added the details in the manuscript.

LINE: 171-179

The sample size calculation was performed based on the previous data [5]. Difference between two independent means and correlation between the groups used to calculate an effect size of 0.31, statistical power of 0.95, and significance level of 0.05. A total sample size of 146 was determined to be adequate.

10) Some of the notations in the text may be spared, to make it more readable. Please check and keep only relevant ones.

Comment 10. Thank you for the kind indication. We revised our manuscript to gain the readability. Moreover, we had used editing service to collect.

11) Very good and well-thought Discussion section, highlighting both novelty and limitations.

Comment. Thank you for the novel indication. We have updated the ideas in the discussion and limitation section.

LINE: 350-357

Fourth, only moderate correlation was observed between subjective SEC analysis and objective AS-OCT analysis (Table 4). Correlation coefficient has similar value in conventional slit-lamp and SEC (r=0.609, and r=0.641). Therefore, it is considerable to increase accuracy, however future studies with more numbers of the cases are needed to gain its precision. Despite these limitations, no other study has demonstrated the utility of a portable and recordable slit-lamp device that utilized only smartphone-based functions. However, prospective studies are needed to verify the effect of the patient’s age, gender, and race.

LINE: 358-376

The availability of ophthalmological examination is limited, especially in developing countries, owing to the lack of expensive medical instruments. Moreover, portable and light weight tools are available in disaster medicine, isolated areas and visiting care. The SEC is not only portable and recordable but is also easy and inexpensive to manufacture and works in conjunction with a smartphone. Therefore, the SEC has potential as a screening tool for ACD and angle estimation that can improve the availably and enlighten the importance of the ophthalmological examination worldwide. For example, in the population attributable risk percentage (PAR%) which addresses the problem of whether a disease is a public health problem [30], primary angle closure has a three times higher PAR% compared to primary open angle glaucoma [31]. Moreover, wireless smartphones are connected globally, which makes the SEC compatible with telemedicine and remote medicine. Furthermore, ophthalmological examinations are based on imaging diagnosis. Thus, artificial intelligence-based diagnosis facilitated by big data collection could be feasible in the future. Our study demonstrated the utility of a new slit-lamp device, i.e., the SEC, for measuring the ACD and estimating the anterior chamber angle compared to the conventional subjective and objective methods. However, the study design had some limitations, such as the lack of gonioscopic evaluation and inclusion of patients from only one ethnicity in the study population. Hence, further prospective studies are necessary to overcome the limitations of the current study.

Reviewer 2 Report

The research article entitled “A study validating the estimation of anterior chamber depth and iridocorneal angle with portable and non-portable slit- 3 lamp microscopy” compares the effectiveness of portable Smart Eye Camera (SEC) for the assessment of anterior chamber depth (ACD) and iridocorneal angle. The data collected by the instrument is compared with conventional slit-lamp microscopy and anterior-segment optical coherence tomography analysis.

Results indicate that SEC might be a promising alternative to conventional slit-lamp assessment. However, neither of them can provide objective analysis that can be obtained from a gonioscopy or OCT for anterior angle assessments. This is also stated in the manuscript.

There are minor suggestions to be considered by the authors.

  • Data availability statement is given in the manuscript. However, the raw data (not the patient names and details) used for mean value calculations should be provided.  
  • It’s found that only moderate correlation can be provided with SEC compared to OCT analysis. What would be done to improve the instrument sensitivity? This should be discussed in the manuscript.

Author Response

Reviewer 2

"The research article entitled “A study validating the estimation of anterior chamber depth and iridocorneal angle with portable and non-portable slit- 3 lamp microscopy” compares the effectiveness of portable Smart Eye Camera (SEC) for the assessment of anterior chamber depth (ACD) and iridocorneal angle. The data collected by the instrument is compared with conventional slit-lamp microscopy and anterior-segment optical coherence tomography analysis.

Results indicate that SEC might be a promising alternative to conventional slit-lamp assessment. However, neither of them can provide objective analysis that can be obtained from a gonioscopy or OCT for anterior angle assessments. This is also stated in the manuscript

Comment. Thank you for the various novel comments. We appreciate the reviewer for the kind instructions. Specific comment and changes are written below.

There are minor suggestions to be considered by the authors.

Data availability statement is given in the manuscript. However, the raw data (not the patient names and details) used for mean value calculations should be provided.

Comment. Thank you for the novel indication. The raw data was downloaded into csv file with the raw value which was round off to fourth decimal places. We revised Data Availability Statement.

LINE: 387-388

Data Availability Statement: All data associated with this study including raw data can be found on the data server in the Department of Ophthalmology, Keio University School of Medicine.

It’s found that only moderate correlation can be provided with SEC compared to OCT analysis. What would be done to improve the instrument sensitivity? This should be discussed in the manuscript."

Comment. Thank you for the novel indication. We have updated the ideas in the limitation section.

LINE: 350-354

Fourth, only moderate correlation was observed between subjective SEC analysis and objective AS-OCT analysis (Table 4). Correlation coefficient has similar value in conventional slit-lamp and SEC (r=0.609, and r=0.641). Therefore, it is considerable to increase accuracy, however future studies with more numbers of the cases are needed to gain its precision.

Reviewer 3 Report

Van herick angle evaluation is just an angle estimation, due to plateau iris configuration, so angle subjective evaluation as described in the manuscript “The depth of the anterior chamber was subjectively evaluated from each conventional slit-lamp and SEC images from the magnified, 3D images structed by the slit-light beam” at the corneal center has no significance in clinical practice. According to EGS guidelines, angle should be screened with van herick grading system, and when in doubt the angle should be analyzed with 4 mirrors or goldmann lens, according to Spaeth systems or the more commonly used Shaffer system.

The study should compare the Van herick obtained from slit lamp and SEC and then compared to AS-OCT findings.

In understand that SEC has a technical limitation and could not tilt the light beam, however overcoming this issue SEC could revolutionize ophthalmic examination, especially in developing countries, as suggested in the last paragraph.

It is retrospective study, however not knowing the ambient luminance value could alter the results, stating this sentence is too simplistic “Both tools were used in the same dark examination area.”

Please explain The Van herick plus grading and the differences with van herick grading.

Please add an imaging to explain data obtained with Casia, not everyone is familiar with AS-OCT (TIA and ACD).

Author Response

Reviewer 3

Van herick angle evaluation is just an angle estimation, due to plateau iris configuration, so angle subjective evaluation as described in the manuscript “The depth of the anterior chamber was subjectively evaluated from each conventional slit-lamp and SEC images from the magnified, 3D images structed by the slit-light beam” at the corneal center has no significance in clinical practice. According to EGS guidelines, angle should be screened with van herick grading system, and when in doubt the angle should be analyzed with 4 mirrors or goldmann lens, according to Spaeth systems or the more commonly used Shaffer system. The study should compare the Van herick obtained from slit lamp and SEC and then compared to AS-OCT findings. In understand that SEC has a technical limitation and could not tilt the light beam, however overcoming this issue SEC could revolutionize ophthalmic examination, especially in developing countries, as suggested in the last paragraph.

Comment. Thank you for the various novel comments. We appreciate the reviewer for the kind instructions. Specific comment and changes are written below.

It is retrospective study, however not knowing the ambient luminance value could alter the results, stating this sentence is too simplistic “Both tools were used in the same dark examination area.”

Comment. Thank you for the novel indication. It is extremely important to know the ambient luminance value where the images were taken. Therefore, we added the details.

LINE: 132-135

However, the recording conditions (e.g., brightness, temperature, humidity, etc.) were maintained consistently to prevent the influence of any environmental factors. Both tools were used in the same dark examination area (30-75 lux. JIS  Z 9110-2010: General rules of recommended lighting levels).

Please explain The Van herick plus grading and the differences with van herick grading.

 Comment. Thank you for the novel indication. It is extremely important to know the ambient luminance value where the images were taken. Therefore, we added the details.

LINE: 137-144

Van-Herick Plus grading was also performed subjectively focusing on the inferior corneal angle image. It is the modified criteria for the golden standard “Van-Herick grading” method for both peripheral ACD and angle assessment [15]. Van-Herick grading uses nasal or temporal (0 or 180 degree) images. Controversially, Van-Herick Plus grading uses inferior (270 degree) angle. Inferior peripheral ACD to peripheral corneal thickness ratio was estimated from the magnified photos. The subjects were divided into four groups to judge the peripheral angle: 1 (<1/4), 2 (1/4 to 1/2), 3 (>1/2 to 1) and 4 (>1) [15].

Please add an imaging to explain data obtained with Casia, not everyone is familiar with AS-OCT (TIA and ACD).

Comment. Thank you for the novel indication. We added figure for the explanation.

LINE: 173-179

Figure 2. Scan of the analysis interface in AS-OCT (CASIA2 Advance system: Tomey, Nagoya, Japan). (A) (B) ACD was define the distance between the apex of the posterior corneal to apex of the crystalline lens using AS-OCT [16,17]. (C) TIA was measured with the apex in the iris angle recess and the arms of the angle passing through a point on the trabecular meshwork 500 μm from the SS and the point on the iris.

Round 2

Reviewer 1 Report

The paper was corrected according to all the comments. In the opinion of this reviewer, the manuscript can be considered for publication in Sensors in the present form.

Author Response

Reviewer 1

The paper was corrected according to all the comments. In the opinion of this reviewer, the manuscript can be considered for publication in Sensors in the present form.

Comment. Thank you for the novel comment. We appreciate the reviewer for the kind instruction. We will reform our MS according to the form of Sensors.

Reviewer 3 Report

You've answered to all question, however I think that the clinical importance of this manuscript is limited, due to angle evaluation technique described. In my opinion, you should add an angle evaluation with a 4 mirrors lens according to Spaeth or Shaffer systems and find a technical solution to tilt the SEC beam and evaluate the angle with van herick technique with SEC.

Author Response

Reviewer 3

You've answered to all question, however I think that the clinical importance of this manuscript is limited, due to angle evaluation technique described. In my opinion, you should add an angle evaluation with a 4 mirrors lens according to Spaeth or Shaffer systems and find a technical solution to tilt the SEC beam and evaluate the angle with van herick technique with SEC.

Comment. Thank you for the novel comment. We appreciate the reviewer for the kind instruction. We deeply understand the importance of the angle evaluation with a 3 or 4 mirror lens of Shaffer system evaluations. However, as the nature of the retrospective approach, it is difficult to add the data of Shaffer system evaluations. Therefore, we emphasis this limitation, add the references in order to lead to the next study.

LINE 324-339

Our study has several limitations. First, although we did confirm the subjective findings obtained with slit-lamp photography, we did not perform any gonioscopic evaluation using three or four mirror goniolens. Gonioscopy is the most traditional method used to estimate the angle width using a gonioscope and slit-lamp microscope [28]. Moreover, Scheie, Shaffer, and Spaeth grading system using a gonioscope are currently the gold standard for the diagnosis and classification of angle evaluation in clinical settings [18, 29, 30, 31]. In the current study, we modified the angle evaluation method using the Van-Herick Plus grading, which is reported to be a simple and relatively accurate technique for both ACD and angle assessment [18]. Moreover, the Van-Herick Plus grade showed a correlation with the anterior chamber angle estimated by AS-OCT [18]. Furthermore, other studies have indicated that AS-OCT measurements are highly correlated with the use of gonioscopy and may possibly be used to assess angle closure risk using non-contact methods [32]. The Van-Herick Plus grading system is able to mimic the utility, accuracy, or value of the gonioscopy method; however, it cannot replace current grading system by gonioscope as the definitive method for diagnosis. Therefore, further study is needed to directly evaluating iridocorneal angle using gonioscope and SEC.
